

# Tuning of controller parameters using Pythagorean fuzzy similarity measure for stable and time delayed unstable plants

Murat Akdağ and Mehmet Serhat Can

Faculty of Engineering and Architecture, Department of Electrical-Electronic Engineering, Tokat Gaziosmanpasa University, Tokat, Turkey

## ABSTRACT

This paper proposes a tuning method based on the Pythagorean fuzzy similarity measure and multi-criteria decision-making to determine the most suitable controller parameters for Fractional-order Proportional Integral Derivative (FOPID) and Integer-order Proportional Integral-Proportional Derivative (PI-PD) controllers. Due to the power of the Pythagorean fuzzy approach to evaluate a phenomenon with two memberships known as membership and non-membership, a multi-objective cost function based on the Pythagorean similarity measure is defined. The transient and steady-state properties of the system output were used for the multi-objective cost function. Thus, the determination of the controller parameters was considered a multi-criteria decision-making problem. Ant colony optimization for continuous domains ($ACO_R$) and artificial bee colony (ABC) optimization are utilized to minimize multi-objective cost functions. The proposed method in the study was applied to three different systems: a second-order non-minimum phase stable system, a first-order unstable system with time delay, and a fractional-order unstable system with time delay, to validate its effectiveness. The cost function utilized in the proposed method is compared with the performance measures widely used in the literature based on the integral of the error, such as IAE (Integral Absolute Error), ITAE (Integral Time Absolute Error), ISE (Integral Square Error), and ITSE (Integral Time Square Error). The proposed method provides a more effective control performance by improving the system response characteristics compared to other cost functions. With the proposed method, the undershoot rate could be significantly reduced in the non-minimum phase system. In the other two systems, significant improvements were achieved compared to other methods by reducing the overshoot rate and oscillation. The proposed method does not require knowing the mathematical model of the system and offers a solution that does not require complex calculations. The proposed method can be used alone. Or it can be used as a second and fine-tuning method after a tuning process.

# INTRODUCTION

In control systems, it is aimed to bring the output variable of the controlled system to a desired reference value. It can analyze the relationship between the system output and the

Corresponding author
Mehmet Serhat Can,
mehmetserhat.can@gop.edu.tr

reference value given in two categories as transient response and steady-state. Transient state features are undershot, rise time, peak time, overshoot, settling time, and the steady-state error value is a steady-state feature. The main aim is to reach this reference value with minimum transient and steady-state values. In some applications, a reference curve is defined and the output is requested to behave by this model. The difference between the output variable of the system and the target reference value is the control error variable ($e$). An approximation of $e$ to zero to satisfy the criteria mentioned above can be accomplished with the unit called a controller. The determination of controller parameters according to the dynamics of the controlled system and the requirements of the application is known tuning procedure. The main problem of the tuning procedure is aimed at achieving as much as possible the transient response, steady-state performance, stability, and robustness against external disturbances. For the tuning of controller parameters, the methods of Ziegler–Nichols, Cohen-Coon, and Åström-Hägglund are crude tuning methods, but they have been the basis for much research in the literature. Today, tuning methods can be classified into two main categories: analytical and iterative optimization (or evolutionary algorithm) techniques. There have been many studies on analytical methods in recent years (*Ma & Chen, 2019*; *Das, Halder & Gupta, 2020*; *Rao, Santosh & Dhanya Ram, 2020*; *Ma et al., 2022*).

Iterative optimization methods are algorithms that are mostly based on the ability of living things to come together and solve a problem they encounter. Popular optimization methods are Genetic Algorithm (GA), Particle Swarm Optimization (PSO), Artificial Bee Colony (ABC), Chaotic Ant Swarm (CAS), Ant Colony Optimization for Continuous Domains ($ACO_R$), Differential Evolution (DE), Cuckoo Search (CS), Penguins Search Optimization Algorithm (PeSOA), Rat Swarm Optimizer (RSO), or a combination of these hybrid algorithms. ABC is a general-purpose optimization algorithm and is generally effective over a wide variety of problems. The ability to converge quickly to the global optimum is a significant advantage of ABC. The algorithm is simple and easy to understand. Another advantage of ABC is that the necessity of parameter tuning is less compared to other algorithms. However, the performance of ABC may decrease when the problem size is very large. In complex search spaces with many local optima, ABC can often fall into local optima. ABC is generally slower compared to faster algorithms (*Anam, 2017*; *Gao, Ye & Wu, 2017*). The Ant Colony Optimization for Continuous Domains ($ACO_R$) algorithm provides a significant competitive advantage when compared with other metaheuristic techniques, thanks to its flexibility and probability-based learning methods. The performance of $ACO_R$ can be adjusted to exhibit more robustness or higher efficiency depending on the user's needs. Moreover, $ACO_R$ provides superiority over direct search methods in problems with a large number of local optima. All these features significantly enhance $ACO_R$'s potential to solve complex optimization problems. Weak points of $ACO_R$ include its performance being sensitive to specific parameter settings, which can complicate the implementation and adaptation of $ACO_R$. Additionally, $ACO_R$ may consume more memory compared to other algorithms, which can be a problem in large-scale problems (*Socha, 2009*; *Socha & Dorigo, 2008*). There are many studies on optimization studies in the literature, the studies presented here are just a few of them in

recent years (*Chaki & Bose, 2022*; *Dhiman et al., 2021*; *Mzili, Mzili & Riffi, 2023*; *Mzili et al., 2022*).

In optimization methods, it is aimed to minimize or maximize a function called objective or cost. For tuning of controller parameters, it is usually aimed to minimize IAE (Integral Absolute Error), ITAE (Integral Time Absolute Error), ISE (Integral Square Error), and ITSE (Integral Time Square Error) performance criteria in the cost function (*Khubalkar et al., 2018*). Here "Error" is the control error variable ($e$). Although the cost function can be reduced to a good degree in the optimization based on the error variable $e$ only, it is not possible to obtain the desired transient response and steady-state performance characteristics together. For example, although a short rise time can be achieved, a high overshoot rate may occur. Even a continuous oscillation may occur, which is undesirable. The cost function can be set up depending on more than one parameter (multi-objective optimization) to get out of the high overshoot rate, and continuous oscillation state, and obtain the minimum steady-state error. A target reference output can be determined according to the transient and steady-state characteristics determined by considering the constraints of the controlled system. Obtaining these reference values can make it possible to provide the desired system output. Thus, almost all (or considered important) transient and permanent state properties can be obtained. In some applications, as multi-objective optimization, a cost function is defined based on the transient and steady-state characteristics of the controlled system, such as rise time, peak time, overshoot, settling time, and steady-state error values along with performance criteria, and the cost function is desired to be minimized (*Şahin & Ayas, 2019*).

Multi-criteria decision making (MCDM) defines the situation where there is more than one criterion and a decision requires taking these criteria into account. In the literature, there are many MCDM methods derived for different applications. These are, linear programming technique for multidimensional analysis of preference (LINMAP), the analytic hierarchy process (AHP), the multi-attribute utility theory (MAUT), and the fuzzy technique for order of preference by similarity to an ideal solution (TOPSIS). The fuzzy set theory-based similarity measure is an effective tool used to measure the overlap between two fuzzy sets. It tells the degree of similarity between two fuzzy sets formed according to some criteria. Often create an ideal set that is most representative of the specified criteria. An instantaneous fuzzy set is created from the measurement data for the criteria sought from the examined system. Then the similarity between these two sets is measured. As a result, the higher the similarity between two sets, the higher the similarity measure will be. From this point of view, cost functions based on similarity measures are created. There are also applications where fuzzy similarity measure-based multi-objective optimization is handled as MCDM. For example, *Can & Ozguven (2017)* considered tuning as an MCDM process and fuzzified the transient and steady-state parameters of the system. Afterward, they used the neutrosophic similarity measure as a cost function between the fuzzy set consisting of fuzzy values they obtained and an ideal fuzzy set determined according to the control criteria. *Ye (2019)* proposed to use Genetic Algorithm instead of the search algorithm used by *Can & Ozguven (2017)* in their study. *Fu et al. (2021)* proposed a self-tuning PID controller design for AC permanent

magnet synchronous motor based on cosine neutrosophic similarity measure. *Ruan, Ye & Cui (2020)* presented a study in which cosine, tangent, and exponential neutrosophic similarity measure is considered. In these optimization studies based on fuzzy similarity measure-based MCDM, the desired transient response, and steady-state performance characteristics can be obtained together at an optimum level.

In closed-loop control systems, controller structures in which the *e* is handled by proportional, integral, and derivative operators are called conventional proportional–integral–derivative (PID) controllers. Depending on the dynamic behavior of the system to be controlled, these structures could also be proportional (P), proportional integral (PI), or proportional derivative (PD). The fact that the PID controller contains three operation operators makes it possible to design controller structures that are not too difficult. This simplicity is why it is often preferred in many control applications. $K_p, K_i$, and $K_d$ are proportional, integral, and differential gains, respectively; these three parameters determine the performance of the PID controller. The conventional PID controller has limitations in unstable, integrating, and resonant processes (*Tan, 2009*). In particular, it is known that they cannot provide good control performance in non-linear systems, linear and high-order systems with time delay, and complex and uncertain systems with unknown mathematical models (*Golcuk, 2022*). A modified form of the conventional PID controller to improve the control performance for unstable, integrating, and resonant processes is the PI-PD controller. In the PI-PD controller, a PD feedback loop is added to the controlled system. This PD addition can make the open loop unstable system open-loop stable. In the PI-PD controller, four parameters named $K_p, K_i, K_d$, and $K_f$ determine the performance of the controller (*Tan, 2009*; *Onat, 2019*). The integral and derivative operators of the conventional PID controller are integer-order, and it is also called integer-order PID (IOPID). Fractional order PID (FOPID) or $PI^\lambda D^\mu$ controller was obtained by using fractional-order integral and derivative operators in the conventional PID controller, and FOPID was first proposed by *Podlubny (1999a)*. With fractional integral and derivative operators, a fractional $PI^\lambda$-$PD^\mu$ controller is obtained. FOPID controller shows better control performance compared to conventional PID controller due to the addition of fractional integral ($\lambda$) and fractional derivative ($\mu$) parameters which are not available in conventional PID controllers. However, the addition of these two parameters brings difficulties in designing a good controller (*Zeng et al., 2015*).

As it is known, many uncertain situations can arise both in MCDM processes in daily activities and engineering problems. This situation also emerges in the analysis of transient and steady-state properties in control systems. For example, in the process of adjusting the parameters of a controller, very different system output responses can be obtained according to the values of the controller parameters. Out of these almost endless answers, uncertain situations may arise that make it difficult to decide while determining the one closest to the desired transient and permanent state characteristics. Pythagorean fuzzy set (PFS) based on fuzzy set theory and used to express uncertain situations is proposed by *Yager (2014)*. In PFS, a quantity is represented by membership and non-membership degrees. These two values provide very useful information for desired and undesired parameter states. The terms between these two values can also be used to

express uncertainty. Similar to other fuzzy set similarity measure approaches, PFS-based similarity measure has been defined in the literature (*Bryniarska, 2020*).

This study proposes a new tuning method for controller parameters in which the Pythagorean fuzzy similarity measure is used as a cost function. In the study, optimization algorithms CS, GWO, PSO, ABC, and $ACO_R$ were tried, but the targeted transient and steady-state properties with CS, GWO, and PSO could not be obtained as desired. On the other hand, the results obtained with ABC and $ACO_R$ could be obtained at a satisfactory level. For this reason, ABC and $ACO_R$ were preferred in the study. The performance of the proposed method was tested on FOPID and PI-PD controllers, which offer effective control performance. Three different plant models were selected. These are the second-order non-minimum phase system, the first-order time-delayed unstable system, and the fractional-order time-delayed unstable system. The proposed method was compared with cost functions based on IAE, ITAE, ISE, and ITSE performance criteria. According to the results obtained, it is seen that the proposed method improves the transient and steady-state characteristics. The main contribution of the proposed method is summarized below:

The proposed method overcomes the uncertain states that will arise in the determination of the controller parameters with the PF similarity measure, thus enabling the intended system transient and steady-state properties to be obtained at the highest possible rate.

The proposed method is based on the transient and steady-state characteristics of the controlled system. It does not require knowledge of the mathematical model of the system. This frees you from complex analytical processes.

The proposed method can be used alone or as a second and fine-tuning method after tuning according to IAE, ITAE, ISE, and ITSE performance criteria. Thus, it makes it possible to obtain transient and steady-state properties that cannot be obtained with cost functions based on IAE, ITAE, ISE, and ITSE performance criteria.

In this article, firstly, the structures of FOPID and PI-PD controllers have been described. Then, ABC and $ACO_R$ algorithms and fuzzy similarity measure based on Pythagorean fuzzy sets are explained. Finally, to evaluate the effectiveness of these methods, the proposed approach is applied to three different systems, and simulation results are presented.

## MATERIALS & METHODS

Fractional mathematics is a more comprehensive form of classical mathematics in which derivative and integral orders can also take fractional or complex values as well as integer values. For many years, the problem of calculating the 0.5-degree derivative of a function in fractional mathematics has attracted the attention of mathematicians and scientists (*Miller & Ross, 1993*). Even today, there is no single answer to this fundamental question, but numerous definitions for fractional calculus have been proposed. Riemann–Liouville, Caputo, and Grunwald–Letnikov are the most well-known fractional mathematics definitions (*Gupta & Kumar, 2019*). In the following sections, these fractional definitions are presented.

## Grunwald–Letnikov, Caputo, and Riemann–Liouville fractional definitions

The Grunwald–Letnikov derivative of function $f(t)$ with fractional order $p$, here $m$ is an integer value satisfying the condition $m < p < m+1$ and $a$ and $t$ are the limit values, is given as Eq. (1) (*Podlubny, 1999b*):

$$_aD_t^p f(t) = \sum_{k=0}^{m} \frac{f^{(k)}(a)(t-a)^{-p+k}}{\Gamma(-p+k+1)} + \frac{1}{\Gamma(-p+m+1)} \int_a^t (t-\tau)^{m-p} f^{(m+1)}(\tau) d\tau. \tag{1}$$

Here, the Gamma ($\Gamma$) function is defined by Eq. (2):

$$\Gamma(x) = \int_0^\infty t^{x-1} e^{-t} dt, \Re(Z) > 0. \tag{2}$$

The Caputo derivative of function $f(t)$ with fractional order $p$ is given as Eq. (3) (*Shah & Agashe, 2016*):

$$_aD_t^p f(t) = \frac{1}{\Gamma(m-p)} \int_a^t \frac{f^{(m)}(\tau)}{(t-\tau)^{p-m+1}} d\tau. \tag{3}$$

Here, $m-1 < p < m$, $m \in N$, $a$, and $t$ are the limit values of integration.

The Riemann–Liouville derivative of function $f(t)$ with fractional order $p$, here $m$ is an integer value satisfying the condition $m-1 < p < m$, $m \in N$, is given as Eq. (4) (*Shah & Agashe, 2016*):

$$_aD_t^p f(t) = \frac{1}{\Gamma(m-p)} \frac{d^m}{dt^m} \int_a^t \frac{f(\tau)}{(t-\tau)^{p-m+1}} d\tau. \tag{4}$$

## Integer approximation methods for fractional order operators

Fractional calculus has significant advantages in modeling and control system performance, but it brings higher computational complexity than integer order calculations because the fractional derivative operator is a non-local operator that includes all the past values of the function. This means that more memory elements are needed to store all the past values. To overcome this computational complexity, integer approximation methods have been developed to approximate the responses of fractional order elements within limited operating ranges (*Deniz et al., 2020*). Continuous fraction expansion (CFE) (*Khovanskii, 1963*), Outstalup (*Oustaloup et al., 2000*), and Matsuda (*Matsuda & Fujii, 1993*) are examples of well-known integer-order models in the literature.

In the continued fraction expansion method (CFE), $s^\alpha$ is determined by expanding the expression $(1+x)^\alpha$. The expansion of the expression $(1+x)^\alpha$ satisfying the condition $0 < \alpha < 1$ is determined by Eq. (5) (*Zawadzki & Włodarczyk, 2017*):

$$(1+x)^\alpha = \cfrac{1}{1 - \cfrac{\alpha x}{1 + \cfrac{(1+\alpha)x}{2 + \cfrac{(1-\alpha)x}{3 + \cfrac{(2+\alpha)x}{2 + \cfrac{(2-a)x}{5+\ldots}}}}}} \tag{5}$$

The expansion of $(1+x)^\alpha$ can alternatively be expressed by Eq. (6):

$$(1+x)^\alpha = \frac{1}{1-} \frac{\alpha x}{1+} \frac{(1+\alpha)x}{2+} \frac{(1-\alpha)x}{3+} \frac{(2+\alpha)x}{2+} \frac{(2-\alpha)x}{5+\ldots}. \tag{6}$$

Here, the 1st-degree continued fraction expansion of $s^\alpha$ is expressed as Eq. (7) by writing $s-1$ instead of $x$:

$$S^\alpha \cong \frac{(1+\alpha)s+(1-\alpha)}{(1-\alpha)s+(1+\alpha)}. \tag{7}$$

The Oustaloup method gives an integer-order approximation of the fractional-order operator $s^\alpha$ using integer-order transfer functions in a certain frequency range $[\omega l, \omega h]$ in specified lower and upper limits (*Marushchak et al., 2022*). Oustaloup approximation of the fractional-order operator $s^\alpha$ given as Eq. (8):

$$s^\alpha = \left(\frac{\omega_u}{\omega_h}\right)^\alpha \prod_{k=-N}^{k=N} \frac{1+s/\omega'_k}{1+s/\omega_k}. \tag{8}$$

Here, $N$ is the approximation order, $\omega_u = \sqrt{\omega_l \omega_h}$ and $\omega'_k$, $\omega_k$ are the zeros and poles of the equivalent integer order transfer functions, respectively. $\omega'_k$ and $\omega_k$ are given by Eqs. (9) and (10):

$$\omega'_k = \omega_l \left(\frac{\omega_h}{\omega_l}\right)^{(k+N+0.5-0.5\cdot\alpha)/(2N+1)} \tag{9}$$

$$\omega_k = \omega_l \left(\frac{\omega_h}{\omega_l}\right)^{(k+N+0.5+0.5\cdot\alpha)/(2N+1)}. \tag{10}$$

The coefficient of the transfer function of the $2N+1$ degree approximation is expressed by Eq. (11):

$$k_\Pi = \left(\frac{\omega_u}{\omega_h}\right)^\alpha. \tag{11}$$

The integer order Oustaloup approximation of $s^\alpha$ can be expressed by Eq. (12):

$$s^\alpha \cong k_\Pi \prod_{k=-N}^{N} \frac{s+\omega'_k}{s+\omega_k} = k_\Pi \frac{B_n s^n + B_{n-1}s^{n-1}+\ldots+B_1 s+B_0}{A_n s^n + A_{n-1}s^{n-1}+\ldots+A_1 s+A_0}. \tag{12}$$

Based on the CFE method, the Matsuda method approximates the fractional-order operator $s^\alpha$ to integer order by calculating the gain at logarithmically spaced frequencies. Logarithmically spaced frequencies, here $n$ approximation order and $\omega_l$ and $\omega_h$ lower and upper-frequency values, are given as Eq. (13) (*Koseoglu et al., 2021*):

$$\omega_k = \begin{cases} \omega_l \left(\frac{\omega_h}{\omega_l}\right)^{\frac{k-1}{n-1}}, n>1. \\ \omega_h, n=1 \end{cases} \tag{13}$$

The gains corresponding to these frequencies are expressed by Eqs. (14) and Eq. (15) (*Krishna, 2011*):

$$d_o(\omega_k) = \left|G(j\omega_k)\right| \tag{14}$$

$$d_n(\omega_k) = \frac{\omega_k - \omega_{n-1}}{d_{n-1}(\omega_k) - d_{n-1}(\omega_{n-1})}. \tag{15}$$

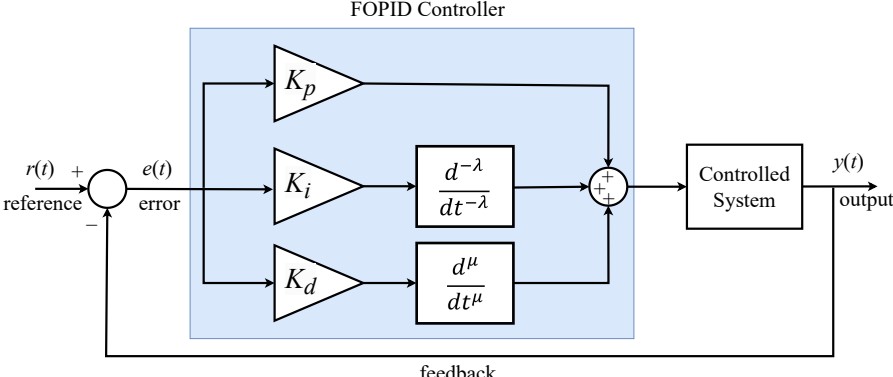

FOPID Controller

**Figure 1  FOPID controller block diagram.**

Then, gain values are placed in a $(N+1)(N+1)$ upper triangular matrix. The obtained upper triangular matrix is expressed as Eq. (16) (*Krishna, 2011*):

$$
D = \begin{bmatrix}
d_0(\omega_0) & d_0(\omega_1) & d_0(\omega_2) & \cdots & d_0(\omega_n) \\
 & d_1(\omega_1) & d_1(\omega_2) & \cdots & d_1(\omega_n) \\
 & & d_2(\omega_2) & \cdots & d_2(\omega_n) \\
\cdots & \cdots & \cdots & \cdots & \cdots \\
\cdots & \cdots & \cdots & \cdots & d_n(\omega_n)
\end{bmatrix}.
\tag{16}
$$

The approximate integer order model's gain coefficients can be expressed as Eq. (17) using the diagonal elements $D_{kk}$ of matrix $D$ (*Krishna, 2011*):

$$
\alpha_k = D_{kk} = \begin{cases}
\dfrac{\left|G(j\omega)\right|}{\omega_k - \omega_{k-1}} & \text{if } k = 0 \\[2mm]
\dfrac{}{d_{k-1}(\omega_k) - d_{k-1}(\omega_{k-1})} & \text{if } k = 1, 2, \ldots, n
\end{cases}
\tag{17}
$$

Then, using the CFE method, the integer degree approximation of the fractional order operator $s^\alpha$ is expressed by Eq. (18).

$$
s^\alpha = \alpha_0 + \cfrac{s - \omega_0}{\alpha_1 + \cfrac{s - \omega_1}{\alpha_2 + \cfrac{s - \omega_2}{\alpha_3 + \cdots}}} = \alpha_0 + \cfrac{s - \omega_0}{\alpha_1 +} \cfrac{s - \omega_1}{\alpha_2 +} \cdots
\tag{18}
$$

## Fractional PID and PI-PD controllers

The fractional PID controller has five parameter values with $\lambda$ (integrator) and $\mu$ (derivative) parameters in addition to the $K_p, K_i, K_d$ parameters that exist in conventional PID. These additional fractional controller parameters allow the controller to operate over a wide range. Also, the FOPID controller is more robust than conventional PID controllers

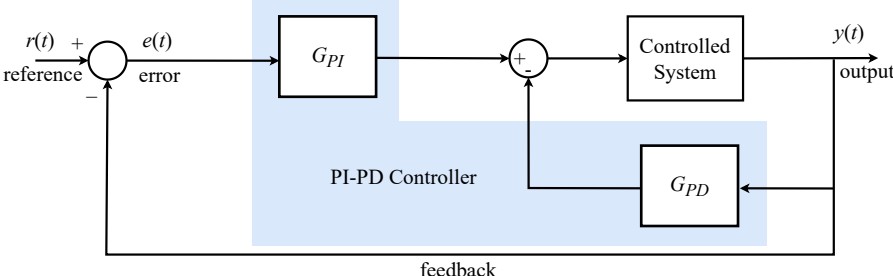

**Figure 2** **PI-PD controller block diagram.**

when the parameters of the system being controlled change (*Agarwal et al., 2019*). The block diagram of the FOPID controller is shown in Fig. 1.

The expressions of the FOPID controller output in the time domain and $s$-plane are given by Eqs. (19) and (20), respectively:

$$u(t) = K_p e(t) + K_i \frac{d^{-\lambda}}{dt^{-\lambda}} e(t) + K_d \frac{d^{\mu}}{dt^{\mu}} e(t) \tag{19}$$

$$u(s) = K_p e(s) + K_i s^{-\lambda} e(s) + K_d s^{\mu} e(s), (\lambda, \mu > 0) \tag{20}$$

In the PI-PD controller, which is obtained by combining a forward PI controller and an internal PD controller with feedback, the PD controller part transforms an open-loop unstable system into an open-loop stable system (*Tan, 2009*), whereas the PI controller part has a significant impact on enhancing the stability process (*Kaya, 2003*). The block diagram of the system including the PI-PD controller is shown in Fig. 2.

Four parameter values are $K_p, K_i, K_d,$ and $K_f$ for the PI–PD controller. The transfer functions of PI-PD controller components are given as Eqs. (21) and (22):

$$C_{PI}(s) = K_p + \frac{K_i}{s} = \frac{K_p s + K_i}{s} \tag{21}$$

$$C_{PD}(s) = K_f + K_d s \tag{22}$$

## Pythagorean fuzzy similarity measure

*Zadeh (1965)* considering the uncertainty in decision-making, proposed the concept of fuzzy sets, and great success has been achieved in various areas involving uncertainty. The fuzzy set $A$, defined in the universal set $X = x_1, x_2, x_3, \ldots,$ can be expressed by Eq. (23):

$$A = \langle x, \mu_A(x) \rangle | x \in X \tag{23}$$

Here, $\mu_A(x)$ is a membership function in the range $[0, 1]$ and is a measure of how compatible an element $x$ is with the fuzzy set $A$. The most common types of membership functions used in fuzzy applications are triangle, trapezoidal, and Gaussian (*Can & Ozguven, 2017*). Researchers have developed different fuzzy set extensions to deal with the increasing fuzziness and uncertainty of real-world data. For this purpose, intuitionistic

fuzzy sets (IFS) were introduced by *Atanassov (1999)* as an extension of the concept of fuzzy sets. The intuitionistic fuzzy set $A$ defined in $X$ is expressed as Eq. (24):

$$A = \langle x, \mu_A(x), \nu_A(x) \rangle | x \in X \tag{24}$$

$\mu_A(x)$ and $\nu_A(x)$ are membership functions showing membership and non-membership degrees, respectively, and take values between $[0, 1]$. Also, the condition $0 \leq \mu_A(x) + \nu_A(x) \leq 1$ must be satisfied and $\pi_A(x)$ (degree of hesitation) is expressed by Eq. (25):

$$\pi_A(x) = 1 - \mu_A(x) - \nu_A(x). \tag{25}$$

There may be cases where the sum of $\mu_A(x) + \nu_A(x)$ in IFS is greater than 1. Pythagorean fuzzy set (PFS) is proposed by *Yager (2014)* to deal with uncertainty satisfying the conditions $\mu_A(x) + \nu_A(x) \leq 1$ or $\mu_A(x) + \nu_A(x) \geq 1$. A Pythagorean fuzzy set defined in $X$ is given as Eq. (26). It must also satisfy the $0 \leq \mu_A^2(x) + \nu_A^2(x) \leq 1$ condition. $\pi_A(x)$ (degree of hesitation) is expressed by Eq. (27) In this way, in decision-making problems; PFS significantly increased the working range compared to IFS (*Bryniarska, 2020*):

$$A = \langle x, \mu_A(x), \nu_A(x) \rangle | x \in X \tag{26}$$

$$\pi_A(x) = \sqrt{1 - \mu_A^2(x) - \nu_A^2(x)}. \tag{27}$$

The similarity measure is used to determine the degree of similarity between two or more sets. There are distance-based, probability-based, fuzzy set theory-based, and graph theory-based approaches to similarity measurement (*Can & Ozguven, 2017*). The similarity measure is usually expressed as a numeric value. It is higher when the data samples are more similar.

$X = x_1, x_2, x_3, \ldots$ is a universal set, $A$ and $B$ are two fuzzy sets defined in $X$, the Pythagorean distance measure between fuzzy sets $A$ and $B$ can be expressed by Eq. (28) (*Peng, 2019*):

$$d(A,B) = \sqrt[p]{\frac{1}{2n(t+1)^P} \sum_{i=1}^{n} \left( \left| (t+1-a)\left(\mu_A^2(x_i) - \mu_B^2(x_i)\right) - a\left(\nu_A^2(x_i) - \nu_B^2(x_i)\right) \right|^P + \left| (t+1-b)\left(\nu_A^2(x_i) - \nu_B^2(x_i)\right) - b\left(\mu_A^2(x_i) - \mu_B^2(x_i)\right) \right|^P \right)} \tag{28}$$

$p$ is the $L_p$ standard and $t$, $a$, and $b$ are parameters that satisfy the condition $a + b \leq t + 1$, $0 < a, b \leq t + 1, t > 0$, and show the uncertainty level. The similarity measure between A and B is expressed by Eq. (29):

$$S(A,B) = 1 - d(A,B). \tag{29}$$

## Optimization methods
### Artificial bee colony algorithm

The artificial bee colony algorithm (ABC), which has been used to solve many optimization problems since its development, is a population-based, heuristic optimization algorithm developed by *Karaboga (2005)* inspired by the food search behaviors of honey bees when collecting nectar from flowers. The ABC algorithm tries to find the maximum

or minimum point of the problem among possible solutions by finding the source with the most nectar (*Karaboga & Akay, 2009*). In the ABC algorithm, the colony consists of three types of artificial bees; employed bees, onlooker bees, and scout bees. The algorithm is initiated by generating food sources, which are distributed randomly in a number equal to the number of employed bees, each corresponding to a solution in the solution space. These food sources, namely solutions, are generated between the randomly determined boundary values ( $l_j$ and $u_j$) as shown in Eq. (30):

$$x_{i,j} = l_j + rand\,(0,1)*(u_j - l_j). \tag{30}$$

Each employed bee is responsible for extracting nectar from only one food source. The quality of each food source is evaluated by the cost function of the solutions. The employed bees look for a new food source in the neighborhood of these food sources. This case can be expressed by Eq. (31):

$$v_{i,j} = x_{i,j} + \varphi_{i,j}*(x_{i,j} - x_{k,j}) \tag{31}$$

$x_k$ represents a randomly selected food source, and $j$ represents a random dimension. The candidate solution obtained as a result of the search conducted in the $x_i$ neighborhood using the random number $\varphi_{i,j}$ is $v_i$. If the quality of the new food source discovered as a result of the search is superior to that of the old food source, the employed bee deletes the old food source information from its memory and records the new food source information. Onlooker bees choose a food source with a probability proportional to the food quality as a result of the information they obtain from the employed bees. If this probability is greater than a randomly generated number, onlooker bees will search for a new food source in the neighborhood of the relevant food source, similar to employed bees (Eq. (31)). If the new food source is of better quality than the old one, it deletes the old one from the memory and records the new food source. When the nectar in a food source is depleted, that is, if the trial number to find a better food source exceeds the specified limit value, the bee responsible for that food source switches to the scout bee phase and performs a random search to discover new sources. The employed, onlooker, and scout bee phases of the algorithm continue until a predefined criterion or the maximum number of cycles is reached, and the algorithm is terminated.

## Ant colony optimization for continuous domains

Ant colony optimization is a population-based metaheuristic optimization algorithm inspired by the foraging behavior of ants (*Dorigo, 1992*). When an ant finds a food source, it evaluates its quality and quantity and carries some of it back to its nest. On its way back to its nest, it leaves a substance called chemical pheromone trails on the road. The amount of pheromone left is related to the quantity or quality of the food found. These trails guide other ants to the same food source. Communication between ants through pheromone trails helps them find the shortest path (*Ojha, Abraham & Snášel, 2015*). Initially developed for discrete optimization problems, the ant colony algorithm was extended to solve continuous optimization problems by *Socha & Dorigo (2008)*. Continuous-time ant colony optimization ($ACO_R$) is a population-based metaheuristic algorithm that iteratively generates solutions.

| $s_1$ | $s_1^1$ | $s_1^2$ | $\cdots$ | $s_1^i$ | $\cdots$ | $s_1^n$ | $f(s_1)$ | $\omega_1$ |
|---|---|---|---|---|---|---|---|---|
| $s_2$ | $s_2^1$ | $s_2^2$ | $\cdots$ | $s_2^i$ | $\cdots$ | $s_2^n$ | $f(s_2)$ | $\omega_2$ |
| | $\vdots$ | $\vdots$ | $\ddots$ | $\vdots$ | $\ddots$ | $\vdots$ | $\vdots$ | $\vdots$ |
| $s_j$ | $s_j^1$ | $s_j^2$ | $\cdots$ | $s_j^i$ | $\cdots$ | $s_j^n$ | $f(s_j)$ | $\omega_j$ |
| | $\vdots$ | $\vdots$ | $\ddots$ | $\vdots$ | $\ddots$ | $\vdots$ | $\vdots$ | $\vdots$ |
| $s_k$ | $s_k^1$ | $s_k^2$ | $\cdots$ | $s_k^i$ | $\cdots$ | $s_k^n$ | $f(s_k)$ | $\omega_k$ |
| | $g^1$ | $g^2$ | | $g^i$ | | $g^n$ | | |

**Figure 3  Solution archive.**

In the $ACO_R$ algorithm, the pheromone information is stored in a table called a solution archive. The solution archive is shown in Fig. 3. The solution archive contains $k$ solutions, each of n dimensions, initially initialized with random values. A solution $s_j$ in the archive represents a solution vector. The $i$th variable of the $j$th solution is denoted by $s_j^i$. $f(s_j)$ represents the value of the cost function. These solutions are added sequentially according to their cost functions as $f(s_1) \leq f(s_2) \leq \ldots \leq f(s_k)$.

In Fig. 3, $\omega$ represents the weight vector. The weight vector $\omega$ is a Gaussian function. The best solution obtains the maximum weight. The weight value $\omega$ of the solution $s_j$ is calculated by Eq. (32). $q$ is a tuning parameter that affects the convergence rate of the algorithm. If it is small, the area around the best-ranked solutions is searched and an early convergence may occur (*Afshar & Madadgar, 2008*):

$$\omega_j = \frac{1}{qk\sqrt{2\pi}} e^{\frac{-(rank(j)-1)^2}{2q^2k^2}}. \tag{32}$$

To select and update a solution in the solution table, the probability of selecting each row in the solution table is calculated given the weight vector $\omega$ (Eq. (33)):

$$p_j = \frac{\omega_j}{\sum_{r=1}^{k} \omega_r}. \tag{33}$$

After the selection is made based on the solution probability value, the parameters of the solution are updated based on the $\sigma$ value given in Eq. (34):

$$\sigma_j^i = \xi \sum_{r=1}^{k} \frac{\left| s_r^i - s_j^i \right|}{k-1}. \tag{34}$$

$\sigma$ is the average distance between the selected solution and the other solutions. $\xi$ is the parameter of the algorithm known as the evaporation rate, and as $\xi$ increases, the convergence of the algorithm decreases. In the final stage of $ACO_R$, $m$ newly generated

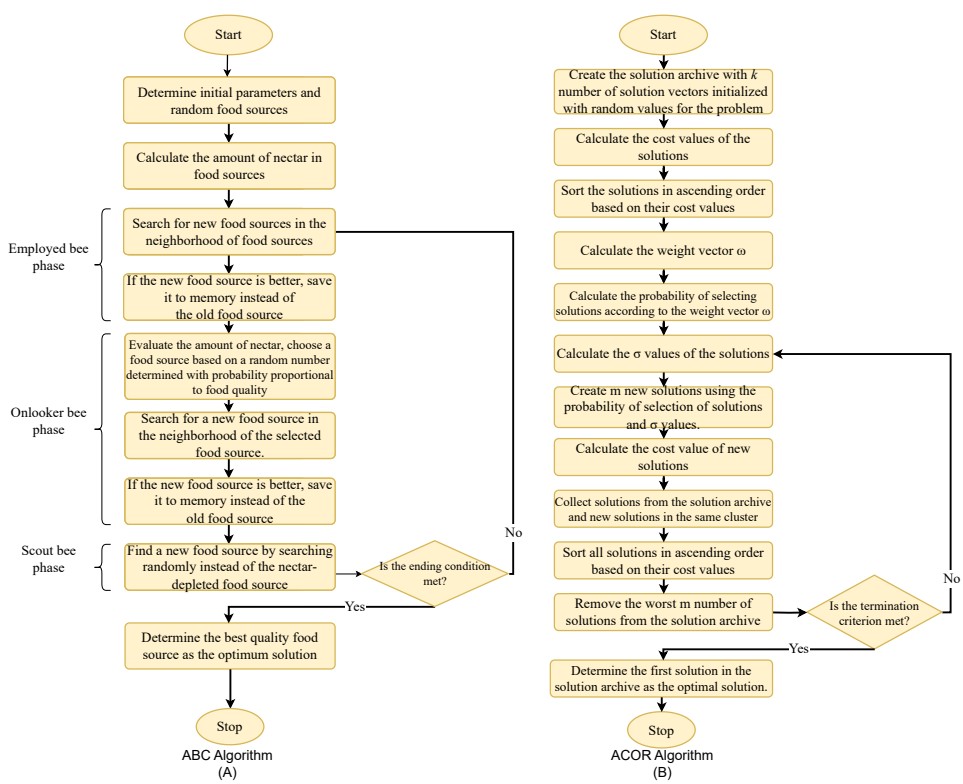

**Figure 4** **Flowchart of the optimization algorithms.** (A) ABC. (B) ACOR.

solutions are added to the initial $k$ solutions. The cost function values of $k + m$ solutions are ranked in ascending order, and in the next step, $m$ worst solutions are removed. Thus, the solution archive is updated with the values of $k$ best solutions. The algorithm is executed until the termination criterion is met and the optimum result is obtained. The flowchart of the ABC and ACO$_R$ algorithm is given in Fig. 4.

## The proposed cost function for tuning of controller parameters

When determining the controller parameters based on the optimization, it usually uses cost functions based on the control error variable $e$. However, $e$'s minimization alone is not enough for its effective performance. In addition to minimizing $e$, the system's transient and steady-state properties (undershoot, rise time, peak time, overshoot, settling time, and steady-state error value) must also be kept at a minimum value. Here, each of the undershoot, rise time, peak time, overshoot, settling time, and steady-state error value values are criteria that must be brought to a certain value. Considering all the values that the controller parameters will take into account, these mentioned parameters will also take a large number of values. In this respect, determining the best system response according to the values of all the transient and steady-state properties of the system is a multi-criteria decision-making problem (*Can & Ozguven, 2017*). There are many different methods for MCDM, but fuzzy similarity measure-based MCDM is a preferred method for the tuning of controller parameters in recent years. In this method, the ideal solution

**Table 1  Some cost functions that are widely used in the literature.**

$$ITAE = \int_0^t t\,|e(t)|\,dt$$

$$ITSE = \int_0^t t e(t)^2 \, dt$$

$$IAE = \int_0^t |e(t)|\,dt$$

$$ISE = \int_0^t e(t)^2 \, dt$$

must be known. The optimum solution is reached by minimizing the distance between the instant solution and the ideal solution.

In the study, the Pythagorean fuzzy similarity measure was used for the information on the distance between the ideal solution and the instantaneous solution, as it allows evaluation the positive and negative distances together. This distance information was defined as a cost function and its minimization was performed with optimization algorithms. GWO, PSO, ABC, and $ACO_R$ optimization algorithms were tried, but the desired results could not be obtained with GWO and PSO. On the other hand, the results obtained with ABC and $ACO_R$ could be obtained at a satisfactory level. For this reason, ABC and $ACO_R$ were preferred in the study. For the ABC algorithm, the number of food sources is 30, and the trial limit parameter is 90. For the ACOR algorithm, the archive size is 30, the sample size is 40, the $q$ parameter value is 0.5, and Pythagorean fuzzy similarity measure parameters are $p = 1$, $a = 1$, $b = 2$, $t = 3$ (*Peng, 2019*). In both optimization algorithms, the number of iterations is 100 and the number of populations is 30.

The most commonly used cost functions in Table 1 were selected to be used in these optimization algorithms, and the cost function obtained with the proposed method was compared with these cost functions. In the equations in Table 1, $e(t)$ represents the difference (error) sign between the reference value applied to the controlled system and the output of the system, and $t$ is the time variable.

The first step in the methodology of the study is to determine the ideal solution for MCDM. The ideal solution is determined by the opinion of an expert. The study, it is aimed to determine the controller parameters that will provide the most suitable transient and steady-state properties for the system output with MCDM. For this reason, the ideal solution is the transient and steady-state properties of the system determined according to expert opinion. Since the Pythagorean fuzzy similarity is used in the study, the ideal solution and the instantaneous solutions that occur during the optimization should be in the Pythagorean fuzzy space. This is done by fuzzification of the real values. Fuzzification is done using Pythagorean fuzzy membership functions for each transient and steady-state feature. The Pythagorean fuzzy similarity between the ideal solution (or fuzzy set) and the fuzzy instantaneous solution (or instantaneous fuzzy set) is calculated and the instantaneous distance is found. With this distance information, a cost function is defined in the study. Minimization of this cost function was performed with ABC and $ACO_R$

optimization algorithms. The parameters found at the end of the optimization are the controller parameters sought. The methodology of the study is described below.

First, based on expert opinion, Pythagorean membership functions are created individually for parameters such as rise time, settling time, overshoot, undershoot, peak time, and steady-state error, *etc.* to satisfy the condition $0 \leq \mu^2(x) + v^2(x) \leq 1$. After, $K_p, K_i, K_d, \lambda, \mu, K_f$ values are initially generated at random. These generated values are provided to the controller selected based on the controller type. Then, the step input is applied to the system, and the transient and steady-state response characteristics (rise time, settling time, overshoot, undershoot, peak time, and steady-state error) are obtained. Each of the obtained system response parameters is passed to membership functions. The real (or instantaneous) Pythagorean fuzzy set $B$, which consists of member and non-member degrees, is the fuzzy set $A$, which consists of targeted ideal values.

$$B_1 = \langle x, \mu_{B-RiseTime}(x), v_{B-RiseTime}(x) \rangle | x \in X$$
$$B_2 = \langle x, \mu_{B-SettlingTime}(x), v_{B-SettlingTime}(x) \rangle | x \in X$$
$$B_3 = \langle x, \mu_{B-Peak}(x), v_{B-Peak}(x) \rangle | x \in X$$

The resulting real Pythagorean fuzzy set vector $B$ is obtained as $B = [B_1, B_2, B_3 \ldots]$. The ideal set vector $A$ is obtained as $A = [A_1, A_2, A_3]$, here $A_1, A_2$, and $A_3$ represent the ideal sets created for each parameter. Here, $A_1 = \langle x, 1, 0 \rangle | x \in X, A_2 = \langle x, 1, 0 \rangle | x \in X, A_3 = \langle x, 1, 0 \rangle | x \in X$. The Pythagorean distance measure between sets $A$ and $B$ is determined by Eq. (28). Then, using the distance measure obtained, the Pythagorean fuzzy similarity measure between these two sets is determined by Eq. (29). In the proposed method, the value of the Pythagorean similarity measure is used as the cost function that optimization algorithms attempt to minimize and cost function is given as Eq. (35):

$$cost = 1 - S(A, B) \tag{35}$$

The optimal $K_p, K_i, K_d, \lambda, \mu, K_f$ values for the controller are determined by running optimization algorithms until the cost value reaches the target value or up to a determined number of iterations. The method of the study is summarized below in the form of processing steps.

*Step 1*: Randomly determine the values of $K_p, K_i, K_d, \lambda, \mu$, and $K_f$ within the specified lower and upper bounds as defined by the optimization algorithm.

*Step 2*: Provide these values to the controller.

*Step 3*: Apply a step input to the system.

*Step 4*: Obtain the response parameters from the system: rise time, settling time, overshoot, undershoot, peak time, steady-state error, *etc.*

Step 5: Each parameter in the response is evaluated using the corresponding membership function from the $B$ vector, and fuzzy sets are obtained for each parameter.

*Step 6*: Calculate the Pythagorean distance between sets $A$ and $B$ using Eq. (28).

*Step 7*: Using this distance, calculate the Pythagorean similarity measure between sets $A$ and $B$ by Eq. (29).

Table 2  Range of values of the membership functions for the rise time.

| Step response parameter | Function ranges | |
| --- | --- | --- |
| | $\mu$(Z-type) | $v$(S-type) |
| Rise time | [1;2] | [1.5;2.5] |

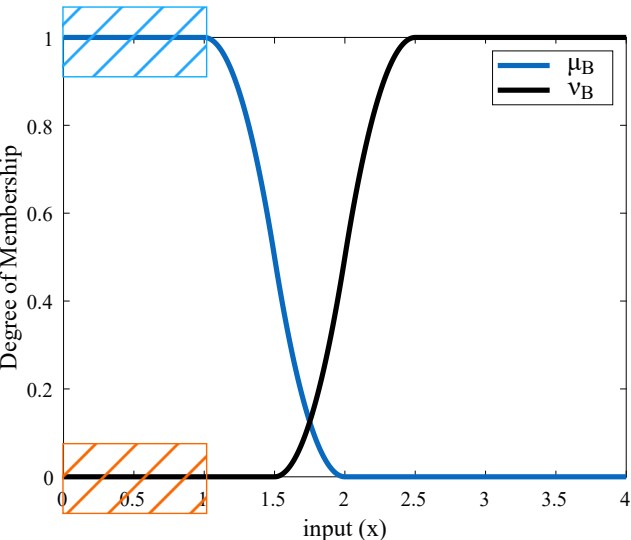

Figure 5  Membership functions of rise time.

*Step 8*: Utilize this similarity measure as a cost function which optimization algorithms aim to minimize.

*Step 9*: Iteratively tune the values of $K_p, K_i, K_d, \lambda, \mu$ starting from *Step 2* until the value of the cost function falls below a pre-determined acceptable target.

Z-type and S-type membership functions generated for an example rise time are shown in Table 2 and Fig. 5. $\mu$ is the Z-type membership function denoting the membership degrees and $v$ is the S-type membership function denoting the non-membership degrees.

For example, if the rise time value is obtained to be 1.8 s after applying the step input to the system, the membership degree of this value to the $\mu$ membership function is calculated as 0.08 and the membership degree to the $v$ membership function is calculated as 0.18 according to the ranges determined in Table 2. In this case, the real fuzzy Pythagorean set $B$ of the rise time is obtained as $B = (0.08, 0.18)$. The membership degree of any value corresponding to the red-orange area in Fig. 5 constitutes the ideal set $A$. These membership degrees also correspond to the blue area. In other words, a rise time value of 1 s or less is the ideal target. In these ranges, the degree of membership of the rise time to the function $\mu$ is 1, the degree of membership to the function $v$ is 0, and the ideal set is shown as $A = (1, 0)$. The similarity measure between sets $A$ and $B$ is calculated by Eq. (29) and is found as 0.3668. In another case, if the rise time value is 1.1 s, the real fuzzy set $B$ is obtained as (0.98.0), and the similarity measure of set $B$ to

**Table 3** Boundary values in optimization algorithms of Example 1.

| Boundary values | Controller parameters | | | | |
|---|---|---|---|---|---|
| | $K_p$ | $K_i$ | $K_d$ | $\mu$ | $\lambda$ |
| Lower bound | 1 | 0.001 | 1 | 0.001 | 0.001 |
| Upper bound | 10 | 1 | 30 | 0.999 | 0.999 |

set $A$ is found as 0.9752. These values are a measure of how similar set $B$ is to set $A$. The closer the value of the similarity measure is to 1, the higher the similarity between the two sets. In this example, for set $B$ to approach the ideal Pythagorean fuzzy set $A$, the incoming rise time value must be less than or close to 1 s. In this case, the real fuzzy set $B$ will be close to $A = (1.0)$. If desired, this process is repeated for other system response characteristics (settling time, overshoot, undershoot, peak, *etc.*), and a total similarity measure is obtained with the help of Eq. (29). This similarity measure value is created a cost function with Eq. (35).

## RESULTS

### Example 1

In this example, a second-order non-minimum phase system is used (*Bingul & Karahan, 2018*). The transfer function of the controlled system is expressed by Eq. (36). This system is difficult to control due to the undershoot and time delay caused by zero in the right half $s$ plane.

$$G(s) = \frac{1 - 5s}{(1 + 10s)(1 + 20s)}. \tag{36}$$

For the control of this system, the 4th-integer Matsuda approximation method is used in the FOPID controller. In the Matsuda approximation, the lower and upper frequencies are chosen as $[10^{-2}, 10^2]$. The lower and upper boundary values used in the ABC and $ACO_R$ algorithms to be used in setting the FOPID controller parameters are shown in Table 3.

Membership functions were created separately for steady-state error, settling time, peak, and undershoot. The range values of the membership functions created for the system are shown in Table 4. The graphical representations corresponding to these range values are given in Fig. 6. It is desirable that the settling time is less than or close to 9 s, the steady-state error is 0.020 or less, the peak value is between 1 and 1.016, and the undershoot value is between 150 and 200.

After optimization, the FOPID controller parameters were obtained, as shown in Table 5, using ITAE, ITSE, IAE, ISE, and the proposed method. The step response characteristics corresponding to the controller parameters are given in Table 6, and the graphical representation of the step response is given in Fig. 7.

The study for Example 1, it is aimed to reduce the undershoot using the proposed method. In addition, the overshoot, settling time, and rise time are desired to be kept as small as possible. Table 6 shows that the IAE cost function stands out among the classical cost functions used in both optimization algorithms. It can be said that the

**Table 4  Range values of membership functions for Example 1.**

| Step response parameters | Function ranges | |
|---|---|---|
| | μ | ν |
| Settling time | [9;11] (*Z*-type) | [10.1;12.1] (*S*-type) |
| Peak | [0;1;1.016;1.026] (Trapezoid) | [1.022;1.032] (*S*-type) |
| Steady-state error | [0.020;0.040] (*Z*-type) | [0.032;0.052] (*S*-type) |
| Undershoot | [0;150;200;210] (Trapezoid) | [206;216] (*S*-type) |

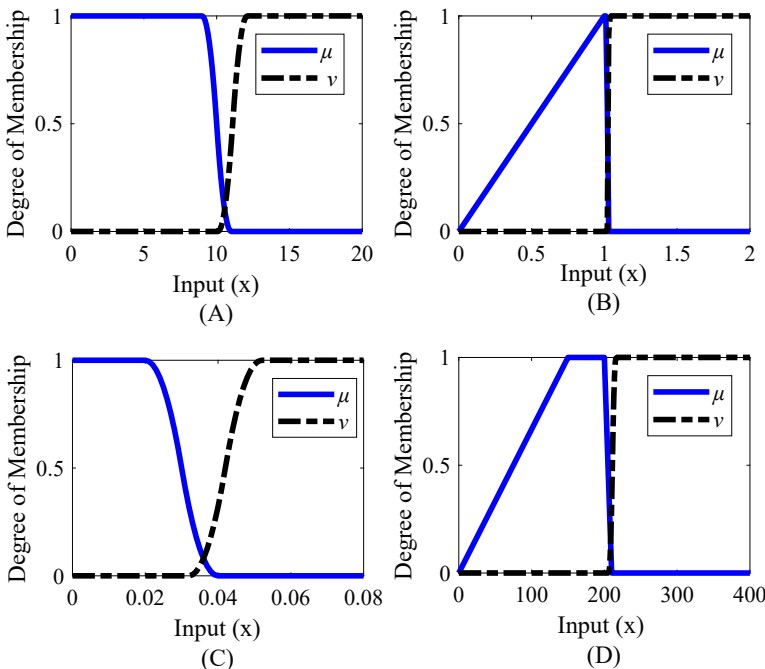

**Figure 6  Membership functions of Example 1.** (A) Settling time. (B) Peak_value. (C) Steady-state error. (D) Undershoot value.

proposed method significantly reduces the undershoot compared to the IAE cost function (156.5210% for ABC and 154.0753% for ACO $_R$). In terms of rise time and settling time, the proposed method is very close to the IAE cost function. Overall, it can be said that the proposed method provides a better control performance.

## Example 2

In this example, a first-order unstable system with time delay is used (*Zheng, Huang & Zhang, 2019*). The transfer function of the controlled system is expressed by Eq. (37). Here is a time delay of 2 s in the system.

$$G(s) = \frac{4}{4s-1}e^{-2s}. \tag{37}$$

**Table 5   FOPID controller parameters of Example 1.**

| Cost functions | Controller parameters | | | | |
|---|---|---|---|---|---|
| | $K_p$ | $K_i$ | $K_d$ | $\mu$ | $\lambda$ |
| Proposed - ABC | 3,85425 | 0,18747 | 26,72912 | 0,94487 | 0,93078 |
| Proposed - ACOR | 3,63808 | 0,22543 | 26,86921 | 0,93238 | 0,88976 |
| IAE - ABC | 4,23500 | 0,19996 | 30 | 0,96348 | 0,94482 |
| IAE - ACOR | 4,35456 | 0,18334 | 29,99040 | 0,96882 | 0,964517 |
| ISE - ABC | 2,96801 | 0,12117 | 20,17954 | 0,99900 | 0,951193 |
| ISE - ACOR | 2,83325 | 0,12908 | 20,13789 | 0,98965 | 0,93637 |
| ITAE - ABC | 4,36656 | 0,14819 | 28,05841 | 0,98312 | 0,99763 |
| ITAE - ACOR | 4,54125 | 0,15282 | 29,99983 | 0,99560 | 0,99899 |
| ITSE - ABC | 4,17118 | 0,16134 | 29,94718 | 0,99850 | 0,97488 |
| ITSE - ACOR | 4,33842 | 0,14845 | 29,99988 | 0,99899 | 0,99513 |

**Table 6   Step response characteristics of Example 1.**

| Cost functions | Step response characteristics | | | | | | |
|---|---|---|---|---|---|---|---|
| | Cost | Rise time | Settling time | Peak | Undershoot | Overshoot | Steady-state error |
| Proposed - ABC | 0,00076 | 2,9122 | 9,07 | 1,00689 | 156,5210 | 0,6894 | 0,0040446 |
| Proposed - ACOR | 0 | 2,6556 | 8,27 | 1,00749 | 154,0753 | 0,7497 | 0,0064680 |
| IAE - ABC | 7,00301 | 2,0965 | 6,77 | 1,00807 | 228,7976 | 0,8072 | 0,0028383 |
| IAE - ACOR | 6,77273 | 2,1943 | 7,11 | 1,00869 | 233,6197 | 0,8697 | 0,001817 |
| ISE - ABC | 9,99346 | 10,6293 | 30,68 | 1,00232 | 100,6296 | 0,2324 | 0,0040789 |
| ISE - ACOR | 9,99504 | 11,2393 | 33,78 | 1,00325 | 94,3124 | 0,3256 | 0,0052995 |
| ITAE - ABC | 17,15853 | 3,3774 | 11,2 | 1,01321 | 201,9418 | 1,3210 | 0,0001257 |
| ITAE - ACOR | 11,44336 | 3,3057 | 12,99 | 1,00212 | 279,07845 | 0,2128 | 5,17395E-05 |
| ITSE - ABC | 11,20865 | 4,6903 | 31,95 | 1,00467 | 288,9481 | 0,4677 | 0,0013975 |
| ITSE - ACOR | 11,08463 | 4,0920 | 43,72 | 1,00057 | 293,1378 | 0,0578 | 0,0002674 |

**Table 7   Boundary values in optimization algorithms of Example 2.**

| Boundary values | Controller parameters | | | |
|---|---|---|---|---|
| | $K_p$ | $K_i$ | $K_d$ | $K_f$ |
| Lower bound | 0,0001 | 0,0001 | 0,0001 | 0,0001 |
| Upper bound | 1 | 1 | 1 | 1 |

PI-PD controller is used to control this system. The lower and upper boundary values used in ABC and ACO$_R$ algorithms are shown in Table 7.

Membership functions were created separately for rise time, settling time, peak, and steady-state error. The range values of the membership functions created for the system are given in Table 8 and the graphical representations corresponding to these range values are given in Fig. 8. According to Table 8, it is desired that the rise time value is between 2.7

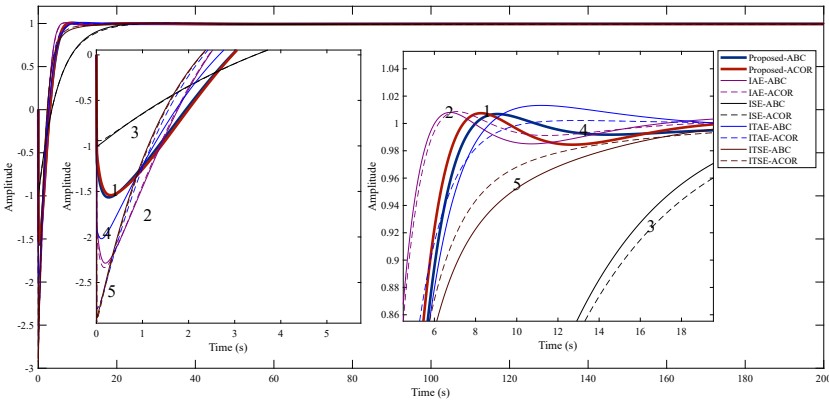

**Figure 7** Step response of Example 1. (1) Proposed method. (2) IAE method. (3) ISE method. (4) ITAE method. (5) ITSE method.

**Table 8** Range values of membership functions for Example 2.

| System response characteristics | Function ranges | |
|---|---|---|
| | μ | ν |
| Rise time | [2.6;2.7;3.2;4.2] (Trapazoid) | [3.8;4.8] (S-type) |
| Settling time | [7;10;12.3;13.5] (Trapazoid) | [13.0;14.2] (S-type) |
| Peak | [0;1;1.004;1.009] (Trapazoid) | [1.0066;1.0116] (S-type) |
| Steady-state error | [0.020;0.040] (Z-type) | [0.032;0.052] (S-type) |

**Table 9** PI-PD controller parameters of Example 2.

| Cost functions | Controller parameters | | | |
|---|---|---|---|---|
| | $K_p$ | $K_i$ | $K_d$ | $K_f$ |
| Proposed -ABC | 0,205123 | 0,009411 | 0,385060 | 0,289589 |
| Proposed - ACOR | 0,170255 | 0,033297 | 0,445509 | 0,391609 |
| IAE - ABC | 0,253132 | 0,003550 | 0,489021 | 0,263644 |
| IAE - ACOR | 0,244017 | 0,011608 | 0,464914 | 0,292575 |
| ISE - ABC | 0,329627 | 0,000100 | 0,588211 | 0,248791 |
| ISE - ACOR | 0,307977 | 0,013471 | 0,579549 | 0,292053 |
| ITAE - ABC | 0,193323 | 0,027191 | 0,478576 | 0,362265 |
| ITAE - ACOR | 0,161996 | 0,037823 | 0,445251 | 0,404665 |
| ITSE - ABC | 0,278962 | 0,000100 | 0,485693 | 0,249870 |
| ITSE - ACOR | 0,255182 | 0,026357 | 0,542944 | 0,339561 |

and 3.2 s, the settling time value is between 10 and 12.3 s, the peak value is between 1 and 1.004, and the steady-state error is below 0.020.

As a result of the optimizations, the PI-PD controller parameters were obtained as in Table 9. The step response characteristics corresponding to these controller parameters are given in Table 10, and Fig. 9 shows the step response graphically.

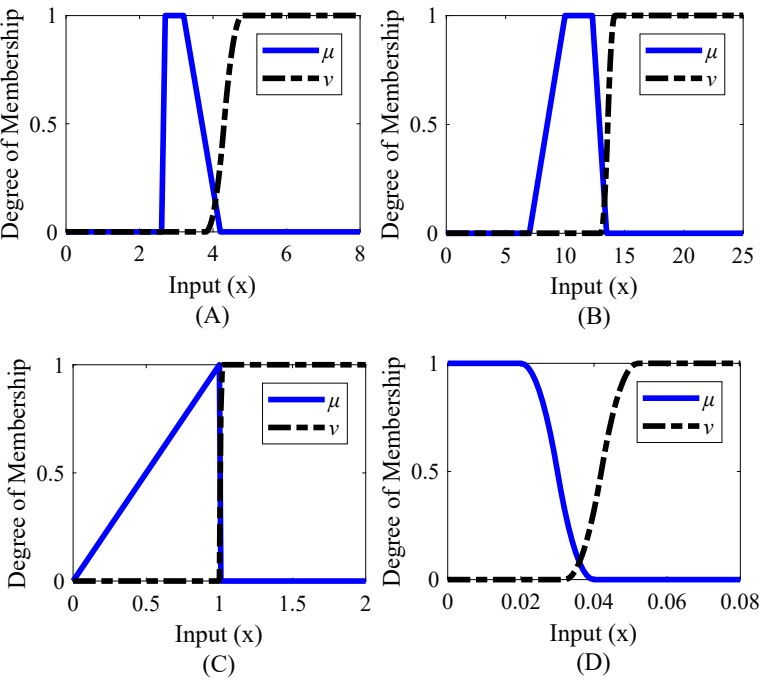

**Figure 8 Membership functions of Example 2.** (A) Rise time. (B) Settling time. (C) Peak value. (D) Steady-state error.

**Table 10 Step response characteristics of Example 2.**

| Cost functions | Step response characteristics | | | | | |
|---|---|---|---|---|---|---|
| | Cost | Rise time | Settling time | Peak | Overshoot | Steady-state error |
| Proposed -ABC | 0 | 3,0527 | 11,499 | 1,00035 | 0,0350 | 0,002161685 |
| Proposed - ACOR | 0 | 3,0344 | 12,247 | 1,00011 | 0,0117 | 5,0806E-05 |
| IAE - ABC | 3,86494 | 2,5420 | 12,153 | 1,00917 | 0,9178 | 0,000226646 |
| IAE - ACOR | 3,85965 | 2,4670 | 10,812 | 1,04549 | 4,5491 | 0,000122512 |
| ISE - ABC | 2,93766 | 1,8340 | 18,051 | 1,14866 | 14,8664 | 0,004685965 |
| ISE - ACOR | 2,97597 | 1,8819 | 18,695 | 1,16246 | 16,2467 | 0,001460794 |
| ITAE - ABC | 10,71917 | 2,8906 | 13,277 | 1,00544 | 0,5446 | 3,00E-06 |
| ITAE - ACOR | 10,52071 | 3,0094 | 11,709 | 1,01983 | 1,9838 | 1,10797E-05 |
| ITSE - ABC | 4,83668 | 2,2584 | 12,186 | 1,06585 | 6,5859 | 0,001661853 |
| ITSE - ACOR | 5,08372 | 2,1716 | 13,576 | 1,10544 | 10,5441 | 0,000542928 |

In Example 2, aimed to reduce the overshoot and keep the settling time and rise time as small as possible with the proposed method. Among the classical cost functions, the lowest overshoot was obtained by using the ITAE cost function in the ABC algorithm (0.5446%). When the proposed method is used in the ABC and ACO $_R$ algorithms, much lower overshoots of 0.0350% and 0.0117% occur, respectively. The duration and amplitude of the oscillation are less when the proposed method is used compared to other

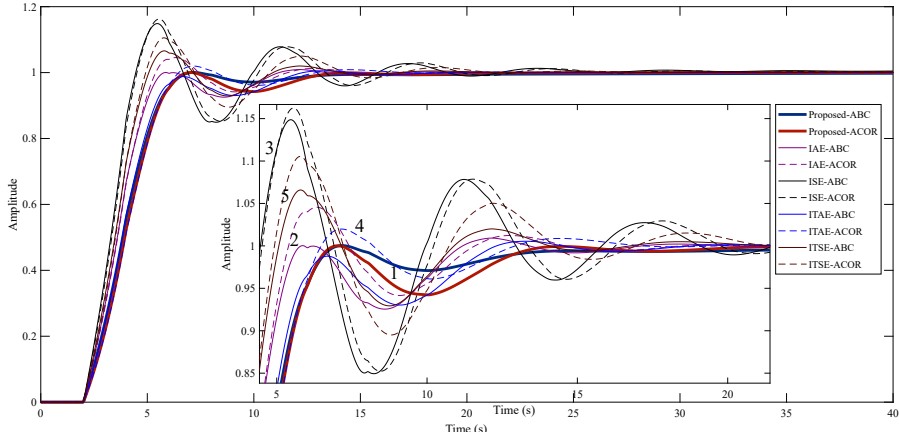

**Figure 9** **Step response of Example 2.** (1) Proposed method. (2) IAE method. (3) ISE method. (4) ITAE method. (5) ITSE method.

**Table 11** **Boundary values in optimization algorithms of Example 3.**

| Boundary values | Controller parameters | | | |
|---|---|---|---|---|
| | $K_p$ | $K_i$ | $K_d$ | $K_f$ |
| Lower bound | 0.01 | 0.01 | 0.01 | 0.01 |
| Upper bound | 5 | 15 | 1 | 10 |

cost functions. In addition, the target values for rise time and settling time are achieved with this method.

## Example 3

In this example, a fractional-order unstable system with time delay is used (*Zheng, Huang & Zhang, 2019*). The transfer function of the controlled system is expressed by Eq. (38). A delay of 0.2 s exists in the system. The Matsuda 4th-integer approximation method was used for fractional-order expressions. In the Matsuda approximation, the lower and upper frequencies are chosen as $[10^{-2}, 10^2]$.

$$G(s) = \frac{1}{s^{1.2} - 1} e^{-0.2s}. \tag{38}$$

PI-PD controller is used to control this system. The lower and upper boundary values used in ABC and ACO$_R$ algorithms are given in Table 11.

Membership functions were created separately for rise time, settling time, peak value, and steady-state error. The range values of the membership functions created for the system are given in Table 12 and the graphical representations corresponding to these range values are given in Fig. 10. It is desired that the rise time value is between 0.2 and 0.3, the settling time value is between 1.2 and 1.4 s, the peak value is between 1.010 and 1.020, and the steady-state error is 0.020 or less.

**Table 12  Range values of membership functions for Example 3.**

| Step response characteristics | Function ranges | |
|---|---|---|
| | μ | ν |
| Rise time | [0;0.20;0.30;0.60] (Trapazoid) | [0.46;0.76] (S-type) |
| Settling time | [0;1.2;1.4;1.6] (Trapazoid) | [1.51;1.71] (S-type) |
| Peak | [0;1.010;1.02;1.03] (Trapazoid) | [1.026;1.036] (S-type) |
| Steady-state error | [0.020;0.040] (Z-type) | [0.032;0.052] (S-type) |

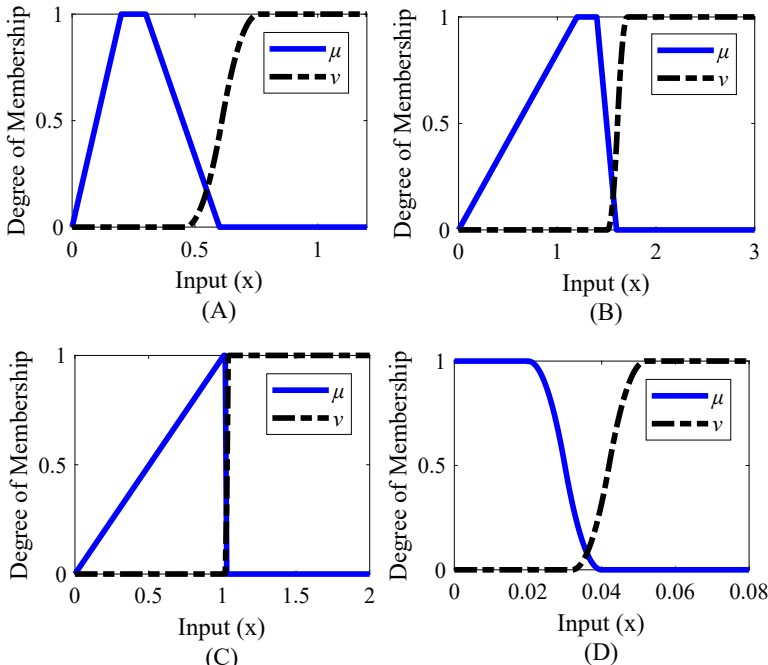

**Figure 10  Membership functions of Example 3.** (A) Rise time. (B) Settling time. (C) Peak value. (D) Steady-state error.

As a result of the optimization, the PI-PD controller parameters were obtained as in Table 13. The step response characteristics corresponding to these controller parameters are shown in Table 14, also with their graphical representation in Fig. 11.

The proposed method is aimed to reduce the overshoot and settling time and to ensure the rapid rise of the system by keeping the rise time at a low value. In the study for Example 3, a high overshoot occurs in classical cost functions. Using the proposed method significantly reduced the overshoot (1.6146% for ABC and 1.9951% for $ACO_R$). When Table 14 is examined, the closest values to the proposed method are obtained with the IAE cost function. However, when this cost function is used, the overshoot and oscillation amplitude remain high. The proposed method is more successful than other cost functions in terms of both settling time and overshoot.

**Table 13  PI-PD controller parameters of Example 3.**

| Cost functions | Controller parameters | | | |
|---|---|---|---|---|
| | $K_p$ | $K_i$ | $K_d$ | $K_f$ |
| Proposed - ABC | 2,90751 | 5,37721 | 0,91313 | 3,59872 |
| Proposed - ACOR | 2,57532 | 7,04771 | 0,96888 | 4,42026 |
| IAE - ABC | 4,18961 | 4,20858 | 0,96295 | 2,56451 |
| IAE - ACOR | 3,41911 | 7,86324 | 0,99990 | 4,02977 |
| ISE - ABC | 5 | 1,81265 | 0,98228 | 1,67829 |
| ISE - ACOR | 4,99999 | 2,00939 | 0,99001 | 1,75439 |
| ITAE - ABC | 2,05523 | 12,17386 | 0,99148 | 6,16731 |
| ITAE - ACOR | 2,35804 | 11,49377 | 0,96658 | 5,72718 |
| ITSE - ABC | 4,92597 | 2,68150 | 0,97324 | 1,94001 |
| ITSE - ACOR | 4,84777 | 3,29170 | 0,98349 | 2,16358 |

**Table 14  Step response characteristics of Example 3.**

| Cost functions | Step response characteristics | | | | | |
|---|---|---|---|---|---|---|
| | Cost | Rise time | Settling time | Peak | Overshoot | Steady-state error |
| Proposed - ABC | 0,06171 | 0,3653 | 1,196 | 1,01614 | 1,6146 | 0,000449084 |
| Proposed - ACOR | 0,07595 | 0,3849 | 1,211 | 1,01995 | 1,9951 | 0,000318058 |
| IAE - ABC | 0,42686 | 0,2349 | 1,347 | 1,06865 | 6,8656 | 0,00064668 |
| IAE - ACOR | 0,43256 | 0,2494 | 1,647 | 1,06372 | 6,3721 | 0,000271386 |
| ISE - ABC | 0,30752 | 0,2076 | 1,32 | 1,11591 | 11,5918 | 0,002759882 |
| ISE - ACOR | 0,30751 | 0,2064 | 1,314 | 1,11929 | 11,9296 | 0,002433699 |
| ITAE - ABC | 0,16289 | 0,2832 | 1,814 | 1,06765 | 6,7650 | 0,000162688 |
| ITAE - ACOR | 0,16284 | 0,2720 | 1,818 | 1,08660 | 8,6604 | 0,000173114 |
| ITSE - ABC | 0,05220 | 0,2056 | 1,333 | 1,13784 | 13,7840 | 0,001456771 |
| ITSE - ACOR | 0,05246 | 0,2058 | 1,547 | 1,140847 | 14,0847 | 0,00103878 |

# CONCLUSIONS

In this article, a tuning method is proposed in which the Pythagorean fuzzy similarity measure is used as a cost function to determine the optimal parameters of the FOPID and PI-PD controllers. To test the proposed method, three previously studied system test systems were utilized: a second-order non-minimum phase system, a first-order unstable system with time delay, and a fractional-order unstable system with time delay. In addition, the performance of the proposed method's cost function is compared to that of widely used time-dependent cost functions such as ITAE, ITSE, IAE, and ISE. ABC and ACO$_R$ algorithms were used for optimization.

When the results obtained from the simulation studies are compared, it is seen that the cost function in the proposed method provides effective control performance by providing improvements in the system response parameters compared to ITAE, ITSE, IAE, and ISE cost functions. With the proposed method, in all three system models examined in the study, the cost function was significantly minimized as intended and thus

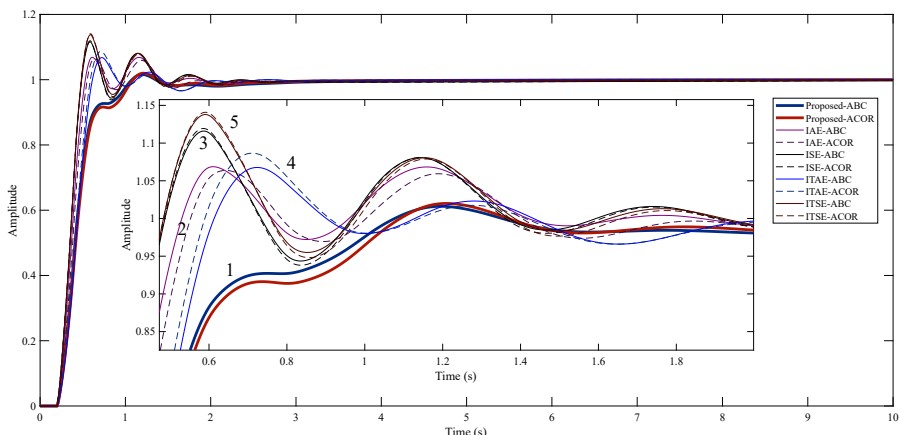

**Figure 11  Step response of Example 3.** (1) Proposed method. (2) IAE method. (3) ISE method. (4) ITAE method. (5) ITSE method.

the targeted transient and steady-state properties can be obtained. In the first example, in a second-order non-minimum phase system, the most important parameter for the control problem is to keep the undershoot ratio low. In addition, other parameters should be reduced together with this parameter. With the proposed method, the undershoot ratio (obtained 156.5210% using ABC and obtained 154.0753% using ACOR) could be reduced together with other parameters. The second example is unstable and has a time delay. In this example, keeping the overshoot rate low and reducing the amount of oscillation comes to the fore. With the recommended method, these two conditions can be improved significantly compared to other methods (overshoot obtained 0.0350% using ABC and overshoot obtained 0.0117% using $ACO_R$). The third example is fractional, unstable, and includes a time delay. In this example, it is important to reduce the overshoot rate (1.6146% using ABC and 1.9951% using $ACO_R$) and oscillation. In all three examples, control error was minimized using other methods, but a large amount of overshoot and oscillation occurred. In the proposed method, control error could be minimized together with other transient and steady-state parameters, as intended.

The proposed method only requires the transient and steady-state properties of the system. It does not require knowledge of the mathematical model of the system. So, the study also provides a solution for tuning the controller parameters that do not require complex calculations. The fact that it does not require complex analytical methods greatly simplifies the tuning process. This offers great convenience in real applications. According to the expected behavior of the system response, the membership functions can be tuned as desired and easily used for different controllers and systems. As a continuation of this study, the proposed method can be tested on reel applications.

### Funding

The authors received no funding for this work.

### Competing Interests

The authors declare there are no competing interests.

### Author Contributions

- Murat Akdağ conceived and designed the experiments, performed the experiments, analyzed the data, performed the computation work, prepared figures and/or tables, and approved the final draft.
- Mehmet Serhat Can conceived and designed the experiments, performed the experiments, analyzed the data, performed the computation work, authored or reviewed drafts of the article, and approved the final draft.

### Data Availability

The MATLAB code is available in the Supplementary File.

### Supplemental Information

Supplemental information for this article can be found online at http://dx.doi.org/10.7717/peerj-cs.1504#supplemental-information.

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
