# Peer review of "Tuning of controller parameters using Pythagorean fuzzy similarity measure for stable and time delayed unstable plants"

_PeerJ Computer Science, doi:10.7717/peerj-cs.1504_

## Round 0.1 · original submission · Major Revisions

Dear authors,

Your paper has been reviewed by three reviewers who asked for revisions of the paper. Please revise the paper according to comments by reviewers, mark all changes in new version of the paper and provide cover letter with replies to them point to point.

Reviewer 1 ·

Basic reporting

See the below comments.

Experimental design

See the below comments.

Validity of the findings

See the below comments.

Additional comments

Thank you for inviting me as a reviewer for this manuscript “Tuning of Controller Parameters using Pythagorean Fuzzy Similarity Measure Based on Multicriteria Decision Making for Stable and Time Delayed Unstable Plants”. The authors have proposed interesting decision making framework based on metaheuristic algorithms (ACO and ABC) . The paper is written well and has enough contribution to be published in the journal. However, hoping to assist the authors in their research efforts, I provide several suggestions for improving the presented work:
. Abstract - The current abstract only describes the general purposes of the article. It should also include the article's main (1) impact and (2) significance on decision making systems. Note that a good abstract should contain aim, methods, findings and recommendations.
. Introduction - You should begin with the problem, the gap, then propose the research question and just after that say what they want to do to address that. Where is the gap? And you should clearly why it is a gap? Once again, if you say that it is a gap, then try to build a case for the gap.
. Why you have used ACO and ABC algorithms for generating functions? Why not other heuristic algorithms like SA, Rat, Grey Wolf etc.? Discuss advantages of these algorithms.
. You should extend the literature review with application of heuristic algorithms and discuss them to show gap. Remove papers published before 2018. I suggest authors to read and discuss below interesting papers: Mzili , T., Riffi , M. E., Mzili, I., & Dhiman, G. (2022). A novel discrete Rat swarm optimization (DRSO) algorithm for solving the traveling salesman problem. Decision Making: Applications in Management and Engineering, 5(2), 287-299.;
Chaki, S., & Bose, D. (2022). Optimisation of spot-welding process using Taguchi based Cuckoo search algorithm. Decision Making: Applications in Management and Engineering, 5(2), 316-328.
Mzili, I., Mzili, T., & Riffi , M. E. (2023). Efficient routing optimization with discrete penguins search algorithm for MTSP. Decision Making: Applications in Management and Engineering, 6(1), 730-743.
. Provide more detail discussion on the results. A discussion section would allow you to come back to your research question and explain once again how their study inform literature in the proposed field in general. I suggest authors to provide some correlation discussion with results obtained based on other relevant algorithms in literature.
. I suggest authors to show benefits and limitations of ACO and ABC algorithms in comparisons to existing used for comparisons. This should be discussed.
. The conclusion section also seems to rush to the end. The authors will have to demonstrate the impact and insights of the research. The authors need to rewrite the entire conclusion section with focus on both impact and insights of the manuscript. Clearly state your unique research contributions in the conclusion section. No bullets should be used in your conclusion section. Provide some future directions.

Reviewer 2 ·

Basic reporting

This study is interesting and has the potential for added value. However, I have some questions for the authors that unfortunately I could not find the answer to these questions in the article.

Does the phrase "...Fuzzy Similarity Measure Based on Multicriteria Decision Making..." in the title refer to fuzzy-based MCDM methods? In such a case, the answer to the following questions should have been clearly emphasized in the study:

(i) Which MCDM method was used in the analyses? (ii) Why weren't the calculation steps of this method shown? (iii) Also, in accordance with the purpose of the study, shouldn't the criteria and alternatives in the first decision matrix be shown in a table?

It is useful to make some sentences meaningful and clear. For example, "fuzzy-based multi-criteria decision making". Please explain the methodology in terms of definition, steps, computational processes, and purposes. No doubt this will be useful for readers.

Experimental design

In my opinion, the methodology has an ambiguous presentation. There is a clear lack of information in this section, which can make it difficult for readers to know and grasp the methodology well.

Please see that the place of "Multi-Criteria Decision Making" in the application is ambiguous for the reader and fix this problem. Make certain sentences precise, meaningful, and understandable.

Validity of the findings

no comment

Additional comments

Presentation of results requires discussion and analysis. A clearer link between the results and the methodology used is needed.

Please review your conclusion with more clarification, removing any ambiguity. The contribution, added value, limitation, and applicability of the approach in the study can be better emphasized.

Reviewer 3 ·

Basic reporting

The paper looks like good material. I like it. But it must be "cleaned" very much and mature a bit more.
Please check and if needed add all the missing punctuation ( , . ; : ) behind the formulas and also lists.
Do not us italics for arbitrary or filling words in formula contexts.
Do the same inside of formulas where punctuation could help the reader / understanding (or is just commonly used).
In fact, inside of the text, sometimes inside of a formula context, commas can help for structuring and reading help.
(Please do not put a . when then a formula comes, You can use a : instead.)
Check all blanks - the ones missing and the ones too much!
Please do not put words like if else otherwise (which you please could use in case studies) in italics (in formula contexts; mind the punctuation there as well.
The same with other words you use in formula contexts; here: italics only if common in mathematics.
Check all blanks - the unnecessary and the still needed. Sometimes blanks are missing behind commas. Be very carefully. The whole team works on it, please.
Please use italics for letters (no other characters, and no full words) in formulas (even tiny ones) whenever missing.
Decent use of italics to enhance the paper's structural appeal and readability.
If you list, e.g., steps ... (with numbers) or so, you can write Step 1 (etc.) in italics, etc.
Let some further experts near you check the whole paper, especially the mathematics. Please.
We do not write Where but here, where or Here ... with sentences and punctuation around adjusted.
Look for a broader view in the Conclusion and Outlook section. Find and see further works by other teams.
Please find and study/compare with works (older and/or newer) on modelling.
If it could be done, it can become a great paper.

Experimental design

no comment

Validity of the findings

no comment

Additional comments

Introduction part and explanations need to be improved.

---

## Round 0.2 · accepted · Accept

Dear authors,

Your paper has been accepted by all reviewers. The title should be changed as the reviewer mentions, but this is the only change required.

Reviewer 1 ·

Basic reporting

The authors have addressed the point of my concern. I am happy with their corrections. Hence, I would like to recommend this manuscript to be published.

Experimental design

The authors have addressed the point of my concern. I am happy with their corrections. Hence, I would like to recommend this manuscript to be published.

Validity of the findings

The authors have addressed the point of my concern. I am happy with their corrections. Hence, I would like to recommend this manuscript to be published.

Additional comments

The authors have addressed the point of my concern. I am happy with their corrections. Hence, I would like to recommend this manuscript to be published.

Reviewer 2 ·

Basic reporting

MCDM methods refer to the methodologies of choosing the most suitable alternative among various alternatives accompanied by certain criteria. As is known, when we look at the literature, we see that there are hundreds of produced MCDM methods. These methods choose not the optimal alternative, but the compromise (or Pareto optimal, according to some) alternative. MCDM methods are not directly related to optimization theory. MCDM theory, by its very nature, does not claim a purely objective selection. Because an evaluation is made on multiple criteria, not on a single criterion. As a matter of fact, there is often some subjectivity for the decision-maker in terms of determining the criteria and weighting the criteria.

When I read the responses to my requests regarding MCDM in the approaches of MCDM algorithms, I realized that the authors had a lack of understanding of "Multi-Criteria Decision Making". When the corrections of the authors are examined carefully, I observe that they confuse both “optimization and MCDM” and “fuzzy numbers and MCDM”. These represent different solutions, concepts, and theories. Fuzzy numbers can be used as well as crisp numbers in MCDM. Fuzzy numbers are not only used in MCDM. In this study, fuzzy numbers are proposed to help decision-making in an optimization problem. Therefore, the study is not based on Multi-Criteria Decision Making. The content of this study is based on optimal decision making based on fuzzy numbers. Therefore, "multi-criteria decision making" should be removed from the content of this study and its title should be changed. Other than that, I have no objections. If the authors think that I have a wrong idea, I invite them to refute this approach.
And I suggest that the final editorial decision on the work be made accordingly.

Experimental design

The content of this study is based on optimal decision making based on fuzzy numbers

Validity of the findings

When the corrections of the authors are examined carefully, I observe that they confuse both “optimization and MCDM” and “fuzzy numbers and MCDM”

Reviewer 3 ·

Basic reporting

The paper is well-improved. I do not have any new suggestions.

Experimental design

Acceptable

Validity of the findings

Satisfactory.

Additional comments

None.